# Determining the feasibility of calculating pancreatic cancer risk scores for people with new-onset diabetes in primary care (DEFEND PRIME): study protocol

Hugh Claridge [ORCID],[1,2] Claire A Price,[1,2] Rofique Ali,[3] Elizabeth A Cooke,[2] Simon de Lusignan [ORCID],[4] Adam Harvey-Sullivan,[3,5] Catherine Hodges,[6] Natalia Khalaf,[7] Dean O'Callaghan,[3] Ali Stunt,[8] Spencer A Thomas,[2] Joanna Thomson,[6] Agnieszka Lemanska [ORCID] [1,2]

**Correspondence to**
Hugh Claridge;
h.claridge@surrey.ac.uk

## ABSTRACT

**Introduction** Worldwide, pancreatic cancer has a poor prognosis. Early diagnosis may improve survival by enabling curative treatment. Statistical and machine learning diagnostic prediction models using risk factors such as patient demographics and blood tests are being developed for clinical use to improve early diagnosis. One example is the Enriching New-onset Diabetes for Pancreatic Cancer (ENDPAC) model, which employs patients' age, blood glucose and weight changes to provide pancreatic cancer risk scores. These values are routinely collected in primary care in the UK. Primary care's central role in cancer diagnosis makes it an ideal setting to implement ENDPAC but it has yet to be used in clinical settings. This study aims to determine the feasibility of applying ENDPAC to data held by UK primary care practices.

**Methods and analysis** This will be a multicentre observational study with a cohort design, determining the feasibility of applying ENDPAC in UK primary care. We will develop software to search, extract and process anonymised data from 20 primary care providers' electronic patient record management systems on participants aged 50+ years, with a glycated haemoglobin (HbA1c) test result of ≥48 mmol/mol (6.5%) and no previous abnormal HbA1c results. Software to calculate ENDPAC scores will be developed, and descriptive statistics used to summarise the cohort's demographics and assess data quality. Findings will inform the development of a future UK clinical trial to test ENDPAC's effectiveness for the early detection of pancreatic cancer.

**Ethics and dissemination** This project has been reviewed by the University of Surrey University Ethics Committee and received a favourable ethical opinion (FHMS 22-23151 EGA). Study findings will be presented at scientific meetings and published in international peer-reviewed journals. Participating primary care practices, clinical leads and policy makers will be provided with summaries of the findings.

## STRENGTHS AND LIMITATIONS OF THIS STUDY

⇒ Early computerisation of UK primary care, incorporating linkage to pathology systems combined with pay-for-performance for chronic disease management including diabetes, helps to ensure population-wide data.

⇒ The extraction software will permit validation of the extracted data by primary care staff prior to transfer to the research team.

⇒ Using glycated haemoglobin results only to define new-onset diabetes means this study is not impacted by the quality of diabetes diagnosis coding in primary care.

⇒ This study will raise awareness of new-onset diabetes' association with pancreatic cancer within the primary care community.

⇒ The study period includes the COVID-19 pandemic; therefore, the data within this period may not reflect the data obtained before or after the pandemic.

## INTRODUCTION

### Pancreatic cancer and early diagnosis

Pancreatic cancer is the seventh leading cause of global cancer deaths, with only 10%–20% of patients diagnosed at a sufficiently early stage for curative intervention.[1 2] Survival can be dramatically improved if diagnosed earlier, at a local rather than distant stage—37% vs 3% 5-year survival rate, respectively.[3 4] However, there are multiple barriers to early diagnosis including the non-specific nature of the early symptoms[5] and lack of suitable diagnostic biomarkers, although advances are being made in this area.[6–12]

As with other health conditions including cancers of other sites,[13] statistical and machine learning clinical prediction models are being developed for clinical use,

to facilitate earlier diagnosis of pancreatic cancer, in particular its most common subtype, pancreatic ductal adenocarcinoma.[3 14 15] These models range widely in complexity,[16–20] with the simplest models including only a few variables that can also be routinely collected in primary care, making them potentially feasible for use in this setting.

### The role of primary care in the UK

In most developed countries, primary care is central to healthcare provision; in the UK, 90% of contacts with the National Health Service (NHS) are through primary care.[21] Primary care providers, including general practitioners (GPs), play a central role in assessing and addressing patients' cancer risk.[22] However, it is estimated that GPs see only one new case of pancreatic cancer every 5 years[23] and, when combined with the non-specific nature of its early symptoms, detection can be very difficult. Clinical diagnostic prediction models are therefore of real potential value for these clinicians, especially given these challenges of diagnosing pancreatic cancer in the context of their busy work schedules.

### Enriching New-Onset Diabetes for Pancreatic Cancer (ENDPAC) model

The simplicity of the ENDPAC model makes it ideally suited for use in primary care as it only uses patient age, weight change and blood glucose measurements, which are routinely collected.[20 24 25] The model is based on the well-documented association of pancreatic cancer with older age and the paradoxical development of diabetes with weight loss.[26–34] It also captures the more rapid onset of glycaemic dysregulation found in pancreatic cancer-related diabetes than found in type 2 diabetes. As the clinical diagnosis of diabetes sometimes occurs months or even years after diabetes onset,[35–37] ENDPAC instead uses the biochemically detected glycaemic onset. This avoids these potential delays thereby maximising new-onset diabetes' potential for the early diagnosis of pancreatic cancer.[20] It therefore means that those deemed by the model as having new-onset diabetes may not otherwise be medically diagnosed as having diabetes, but rather hyperglycaemia. The model has undergone external validation in three separate studies, two using data from the USA and one using data from Israel, establishing that ENDPAC demonstrates a reasonable ability to differentiate patients with type 2 diabetes from those with glycaemia-defined diabetes who later develop pancreatic cancer.[24 25 38] It is for these reasons that this study will investigate ENDPAC's feasibility for use in UK primary care settings.

### ENDPAC scores

ENDPAC calculates risk scores that patients have pancreatic cancer, by using their age and changes over time to their weight and blood glucose results. According to the model's developers, a score ≤0 has a sufficiently high negative predictive value for pancreatic cancer that those with this score can be deemed as only needing management for type 2 diabetes, given their very low risk of pancreatic cancer. A score ≥3 is considered to warrant clinical workup for pancreatic cancer.[20] This is because in the original development study and three subsequent external validation studies, patients with a score ≥3 had, respectively, a 3.6%, 2.0%, 2.6% and 2.6% 3-year risk of pancreatic cancer, with sensitivities of 78%, 63%, 42% and 54%.[20 24 25 38] The reduced performance in the external validation studies is unsurprising, as performance is often lower when models are applied to different populations than those used to build the model.[39] Furthermore, Sharma *et al*[20] suggested that with sufficient additional case review processes, 50% of false positives can be removed, increasing the 3-year risk of pancreatic cancer for patients with a score ≥3 from 3.6% to 10%.

### Data extraction and value selection software

In order to calculate ENDPAC scores, specific results for HbA1c, weight or body mass index (BMI) need to be obtained within defined time periods to quantify the changes in these results over time.[20] The process of manually searching patient record management systems for participants meeting the inclusion criteria and extracting the correct results for every individual would be extremely time-consuming. In addition to this, manually performing this process, or permitting the participating primary care practices to develop their own mechanisms for doing so, would bring into question the reliability and accuracy of the results extracted for each individual as the value selection criteria are extremely complex to apply. This could potentially undermine the validity of the findings of this feasibility study.

To address these issues, we will develop and provide software to primary care practices for the UK's two main primary care patient record management systems which are used by over 85% of primary care providers: EMIS Web by EMIS Health and SystmOne by TPP.[40] The software will search the electronic healthcare records within these patient record management systems, extract the data for those meeting the inclusion criteria and apply complex value selection procedures to obtain the data needed to calculate ENDPAC scores. This will ensure the consistency, reproducibility and reliability of the data extracts, as well as use as little of the practice staff's time as possible, thereby increasing the potential usability of ENDPAC in clinical settings.

### Rationale for this feasibility study

The ENDPAC model was developed using data from the USA.[20] However, to date, ENDPAC has not been reported as tested in the UK. While patient weight and blood glucose measures are routinely collected in many developed nations, there may be different approaches to gathering these data and different units of measurement used

both between and within countries. Therefore, through the development of data extraction, value selection and ENDPAC score calculation software, we will be investigating the availability and quality of these data. This is a critically important step to undertake before considering using ENDPAC in UK clinical practice, as these aspects will directly impact whether ENDPAC scores can be calculated using UK primary care data.

### Study aim and objectives

The aim of this study is to determine the feasibility of calculating ENDPAC scores for people with new-onset diabetes in UK primary care practices.

The objectives are as follows:

1. Develop data extraction and value selection software for primary care. We will work with software developers for primary care patient record management systems to develop and test the software for data extraction and value selection.
2. Extract data from 20 primary care practices and evaluate the availability and quality of data.
3. Develop ENDPAC score calculation software and undertake descriptive data analysis. We will report the number of people with ENDPAC scores warranting referral for pancreatic cancer investigations and their clinical and demographic characteristics.

### METHODS AND ANALYSIS
### Design and setting

Determining the feasibility of calculating pancreatic cancer risk scores for people with new-onset diabetes in primary care (DEFEND PRIME) will be a multicentre observational study with a cohort design in the UK. We will extract anonymised data from 20 primary care practices for people with new-onset diabetes identified within the most recent 3-year period.

### Primary care practice recruitment

We will use several recruitment strategies:
► Presentations and networking at conferences and meetings attended by clinicians and academics working in the early detection field.
► Newsletters from relevant charities and clinical governing bodies.
► Advertising on social media channels.
► Dissemination of study information by stakeholders and colleagues through professional networks.

Practices will enrol by completing a data sharing agreement. Based on an hourly rate of £50 for an estimated 7 hours' work to extract the data, each practice will be reimbursed £350.

### Participant eligibility

Participants must be at least 50 years old and with new-onset diabetes identified in the last 3 years. To be in accordance with ENDPAC, for the purposes of this study, new-onset diabetes will be defined by an abnormal glycaemic test result of HbA1c ≥48 mmol/mol (6.5%). All prior HbA1c test results for the participants must be below this level.

### Data extraction and value selection software

We will develop the data extraction software with software developers who specialise in creating searches in patient record management systems. The software developed and provided to primary care practice staff will include detailed instructions to enable staff to run the data extraction and value selection software.

Table 1 details the data that will be extracted, which is modified from Sharma et al.[20] Prior to transfer to the research team, the value selection software will select the preferred HbA1c, weight and BMI results according to the priorities defined by Sharma et al[20] from the results available at the multiple defined timepoints for these variables shown in table 1, with any excess results removed. The software will be piloted prior to use to ensure accuracy, with spot checks undertaken by selected primary care staff on extracted data.

The final extract file containing anonymised data will then be securely transferred to the University of Surrey and stored on secure research drives accessible only by the research team. Participants will not be identified or contacted during this study. Figure 1 shows the various steps that will be undertaken following the enrolment of a primary care practice in the study, including the running of the data extraction software, value selection software and transfer of the extracted data to the study team.

### ENDPAC score calculation

For participants with the required results, ENDPAC scores will be calculated by the research team using the score calculation software being developed, according to the process defined by ENDPAC's developers. This software will use HbA1c (mmol/mol) results equivalent to the original calculator's fasting blood glucose and estimated average glucose results.[20]

### Data analysis

Table 2 shows the descriptive statistics that will be used to describe the demographics of the cohort, including counts with percentages, means with standard deviations (SD) and medians with interquartile ranges (IQR). We will assume that data are missing at random and therefore results will be calculated using the available data. The proportion of missing data will be described using counts with percentages. We will provide counts with percentages of participants for whom HbA1c, weight, BMI and height results are available and specifying for which timepoints. For the participants meeting the study's inclusion criteria, we will be able to assess the availability of data for the variables listed through the analysis summarised in table 2.

We will report on the number of cases for whom ENDPAC scores can be calculated per practice, in addition to the distribution of the scores into high-risk (≥3), intermediate-risk (2–1) or low-risk (≤0) groups

Table 1   Description of the variables to be extracted at the timepoints specified to enable calculation of ENDPAC scores

| Variable name | Timepoint/time window | Details |
| --- | --- | --- |
| Index HbA1c result (1) | Index date | HbA1c result in mmol/mol for those meeting the inclusion criteria |
| Age group | Index date | Participant's age group at index date, in 5-year bands |
| Gender (m/f) | N/A | Latest recorded |
| Height | N/A | Latest recorded aged 18 years or older |
| HbA1c (2a) | Between 9 and 15 months prior to index date | If multiple values, provide highest value |
| HbA1c (2b) | Between 15 and 18 months prior to index date | If multiple values, provide highest value |
| HbA1c (2c) | Between 6 and 9 months prior to index date | If multiple values, provide highest value |
| HbA1c (2d) | Between 3 and 6 months prior to index date | If multiple values, provide highest value |
| HbA1c (2e) | Between 18 and 24 months prior to index date | If multiple values, provide highest value |
| Weight or BMI and height (1a) | Between index and 3 months after index date | If multiple values, provide earliest value (closest to index date) |
| Weight or BMI and height (1b) | Between index and 3 months prior to index date | If multiple values, provide latest value (closest to index date) |
| Weight or BMI and height (2a) | Between 9 and 12 months prior to index date | If multiple values, provide earliest value (closest to 12 months prior to index date) |
| Weight or BMI and height (2b) | Between 12 and 15 months prior to index date | If multiple values, provide latest value (closest to 12 months prior to index date) |
| Weight or BMI and height (2c) | Between 15 and 18 months prior to index date | If multiple values, provide latest value (closest to 15 months prior to index date) |
| Weight or BMI and height (2d) | Between 3 and 9 months prior to index date | If multiple values, provide earliest value (closest to 9 months prior to index date) |
| Weight or BMI and height (2e) | Between 18 months and 10 years prior to index date | If multiple values, provide latest value (closest to 18 months prior to index date) |
| Pancreatic cancer diagnosis | Earliest result in participant's history | If code for 'Malignant tumour of pancreas (disorder)' or any child* codes (excluding recurrence or metastasis) is present, state whether before or after index date |
| CA 19-9 result | Earliest result in participant's history | State whether below or above threshold value and if before or after index date |
| CEA result | Earliest result in participant's history | State whether below or above threshold value and if before or after index date |

*The term 'child' refers to the relationship between the main code (the 'parent' code) and its related ('child') codes, and does not relate to the age or relationship of the participants.
BMI, body mass index; CA 19-9, cancer antigen 19-9; CEA, carcinoembryonic antigen; HbA1c, glycated haemoglobin.

for pancreatic cancer at the time they first meet the glycaemic definition of new-onset diabetes,[20] and include the timepoints from which the HbA1c and weight results were taken. This will enable us to provide estimates on the number of patients who would need clinical workup for pancreatic cancer if the ENDPAC model were to be deployed across the UK, thereby assessing the potential resource burden on the NHS.

## Project governance
The study will be overseen by a steering group of GPs from the Surrey and Sussex Cancer Alliance. They will meet

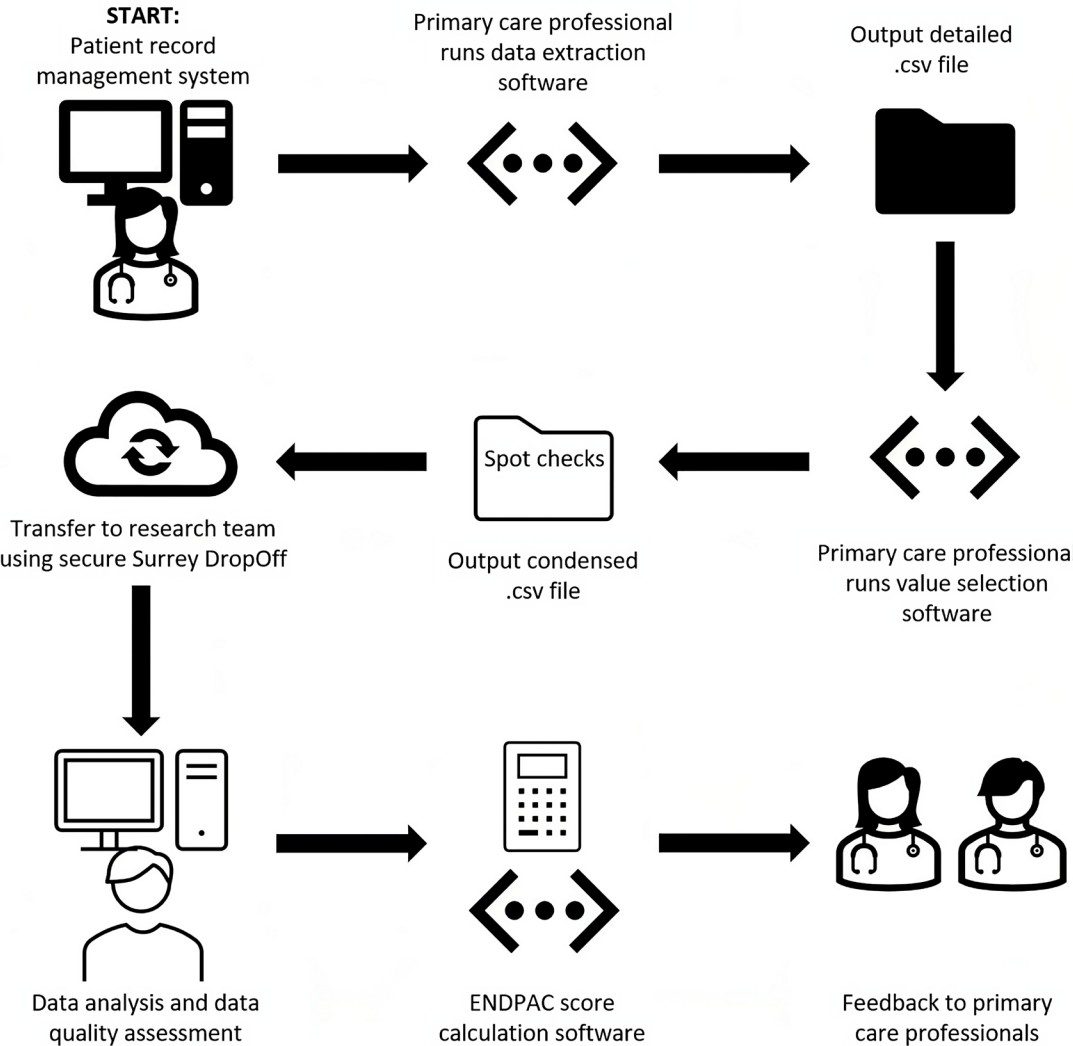

**Figure 1** Flowchart providing an overview of the main steps for the participating primary care practices in the DEFEND PRIME study. ENDPAC, Enriching New-onset Diabetes for Pancreatic Cancer.

with the study team every 2 months to discuss progress. The steering group and other stakeholders, including the NHS Cancer Programme strategy team and the charitable patient advocate group Pancreatic Cancer Action have already been and will continue to be involved throughout the study, providing advice and guidance on study design, recruitment and dissemination strategies.

### Patient and public involvement and engagement

We have been working closely with a pancreatic cancer survivor, who was involved in the conception of the study and this protocol's development. They contribute to the project through the regular 2 monthly project meetings. They are updated on progress and have input on the study's design, conduct and recruitment processes. In the future, they will be involved in dissemination. We are also working with two charities, Pancreatic Cancer Action and Pancreatic Cancer UK, who have well-established patient and public involvement groups, including pancreatic cancer survivors and family members of those

with the disease. We will consult these groups during study delivery and dissemination. Their expertise and feedback will be incorporated throughout this study and they will also support study dissemination, including involvement in conference presentations, webinars and publications, as well as in developing plain English summaries.

### DISCUSSION

This will be the first study the authors are aware of to determine whether ENDPAC scores can be calculated for patients in UK primary care. We will achieve this by developing scalable software that will extract the required data, conduct the complex value selection process and calculate ENDPAC scores. This will enable the assessment of the availability and quality of the data required for score calculation, in addition to enabling primary care practice staff to validate a portion of the extracted data prior to transfer to the research team thus providing confidence

**Table 2** Details of the planned descriptive statistics for the specified variables

| Variable | | Descriptive statistics |
|---|---|---|
| Age group | 5-year bandings: 50–54, 55–59, 60–64 etc | n (%) |
| Gender | Male | n (%) |
| | Female | |
| Body weight (kg), BMI and height | Result at index date | Mean (SD) |
| | Result ~1 year prior to index date | |
| | Weight change | |
| | Weight change | Median (IQR) |
| | Between index date and 3 months after index date | n (%) |
| | Between index date and 3 months prior to index date | |
| | Between 9 and 12 months prior to index date | |
| | Between 12 and 15 months prior to index date | |
| | Between 15 and 18 months prior to index date | |
| | Between 3 and 9 months prior to index date | |
| | Between 18 months and 10 years prior to index date | |
| HbA1c (mmol/mol) | Result at index date | Mean (SD) |
| | Result ~1 year prior to index date | |
| | HbA1c change | |
| | HbA1c change | Median (IQR) |
| | Between 9 and 15 months prior to index date | n (%) |
| | Between 15 and 18 months prior to index date | |
| | Between 6 and 9 months prior to index date | |
| | Between 3 and 6 months prior to index date | |
| | Between 18 and 24 months prior to index date | |
| HbA1c category* at index date | Category 5 | n (%) |
| | Category 4 | |
| HbA1c category ~1 year prior to index date | Category 3 | n (%) |
| | Category 2 | |
| | Category 1 | |
| Change of HbA1c category | | Mean (SD) |
| Change of HbA1c category | | Median (IQR) |
| ENDPAC score | −6 to +11 | n (%) |
| CA 19-9 result | Available | n (%) |
| | Above threshold value | n (%) |
| CEA result | Available | n (%) |
| | Above threshold value | n (%) |
| Pancreatic cancer diagnosis | Available | n (%) |
| | Prior to index date | n (%) |
| | At or post index date | |

*HbA1c categories are defined by the ENDPAC calculator, depending on the HbA1c levels at index date and pre-index date.[30]
BMI, body mass index; CA 19-9, cancer antigen 19-9; CEA, carcinoembryonic antigen; ENDPAC, Enriching New-onset Diabetes for Pancreatic Cancer; HbA1c, glycated haemoglobin; IQR, interquartile range; SD, standard deviation.

in the findings. Through assessing the availability and quality of the data, the feasibility of rolling-out ENDPAC in UK primary care can be established, and the resource impact on the NHS estimated, based on the number of participants warranting clinical workup through sufficiently high ENDPAC scores.

The quality of routine data presents a challenge in any data-driven study. For example, weight, BMI and HbA1c measurements are opportunistically collected in clinical practice, and therefore are not necessarily available at regular time intervals. Through discussion with the study's steering group, even though weight and height

are needed to calculate BMI, patient record management systems do not always require the underlying values at that timepoint to be entered when recording or calculating BMI. As ENDPAC requires weight results for score calculation, we are extracting BMI and height values in addition to weight, to enable back-calculation of weight results if only BMI and height values are provided at the required timepoints. This will maximise the number of eligible participants for whom an ENDPAC score can be calculated. We will provide feedback to the practices if any particular issues are encountered with missing results in their practices, and how they might improve on this.

Given the aims of this feasibility study, no sample size calculation was performed.[41] The chosen sample size is pragmatic and based on the findings of preliminary exploratory analysis undertaken by the research team of coded diabetes diagnoses in primary care records. This indicated that, on average, approximately 30–40 patients per practice in the UK are diagnosed with new-onset diabetes annually. As the study period covers 3 years, and as we are defining new-onset diabetes solely using HbA1c results rather than relying on coded diagnoses, it is estimated that each participating primary care practice will provide data for at least 100 participants. Therefore, with 20 practices participating, approximately 2000 participants' records will be provided and we regard this to be suitable to achieve this feasibility study's aims. Only anonymised data will be extracted and analysed, meaning that individual participants' ENDPAC scores will not be reported.

The study period includes the COVID-19 pandemic, thus the data extracted may not reflect data obtained before or after the pandemic. This is because patients' healthcare-seeking behaviour changed during the pandemic, resulting in atypical fluctuations in attendance to UK healthcare settings, including primary care.[42] Therefore, it may be that fewer HbA1c and weight results will be present in the participants' records in the pandemic period, and this will be taken into consideration.

In this study, we will use single, unpaired HbA1c results, while ENDPAC was originally developed for use with paired results from a combination of fasting blood glucose, random blood glucose, HbA1c or oral glucose load test results.[20] This is because the external validation studies reported that participants had substantially more HbA1c results than other blood glucose measurements and recommended that ENDPAC be applied in a real-world setting to those diagnosed with diabetes through HbA1c only.[24 25] In addition, Khan *et al*'s external validation successfully used single HbA1c results, rather than requiring paired results.[24] Furthermore, the UK stakeholder GPs involved in planning the current study have highlighted that HbA1c is their preferred means of assessing patients' blood glucose and is the principal method for diabetes monitoring in the UK.[43] It is for these reasons that HbA1c results will be used in this study.

The 3-year time window we are using in this study is based on the significantly increased risk of pancreatic cancer in the 3 years after diagnosis of new-onset diabetes.[20 44 45] Sharma *et al*[20] suggested that with sufficient additional case review processes for those having ENDPAC scores calculated, 50% of false positives can be removed, increasing the 3-year risk of pancreatic cancer for patients with an ENDPAC score ≥3 from 3.6% to 10%. This process includes reviewing patients' records for other causes of weight loss, recent steroid use causing rapid blood glucose increases and uncontrolled diabetes causing rapid weight gain pre-index and rapid weight loss post-index. For the purposes of this feasibility study, such additional case review processes are not considered necessary for inclusion within the data extraction process. This is because depending on the outcome of this feasibility study, we plan to design and deliver a clinical intervention collaborating with patients and clinicians, aiming to improve early diagnosis by using ENDPAC scores. In the future study, after clinical consultation involving manual case review by clinicians to assess each participant's suitability for participation, participants with an elevated ENDPAC score will be invited for further investigations, such as blood tests and pancreatic scans, to rule out or diagnose pancreatic cancer.

## ETHICS AND DISSEMINATION

This project has been reviewed by the University of Surrey University Ethics Committee and has received a favourable ethical opinion (FHMS 22-23151 EGA). We will comply with the legal and policy requirements of the University of Surrey.

Data extracts created as part of this project will remain under the management of primary care practices. Data will not be made open access or deposited in any repository, as outlined in the data sharing agreement. Subject to all necessary approvals, data may be made available for secondary use by the primary care practices who remain data controllers.

Results will be presented at scientific meetings and published in international peer-reviewed journals. Summaries will be provided to the participating primary care practices, clinical leads and policy makers.

**Author affiliations**
[1]School of Health Sciences, Faculty of Health and Medical Sciences, University of Surrey, Guildford, UK
[2]National Physical Laboratory, Teddington, UK
[3]Tower Hamlets Network 1 Primary Care Network, London, UK
[4]Nuffield Department of Primary Care Health Sciences, University of Oxford, Oxford, UK
[5]Centre for Primary Care, Wolfson Institute of Population Health, Queen Mary University of London, London, UK
[6]Surrey and Sussex Cancer Alliance, Guildford, UK
[7]Section of Gastroenterology and Hepatology, Department of Medicine, Baylor College of Medicine, Center for Innovations in Quality, Effectiveness, and Safety (IQuESt), Michael E. DeBakey Veterans Affairs Medical Center, Houston, Texas, USA
[8]Pancreatic Cancer Action, Oakhanger, UK

**Acknowledgements** The work of NPL coauthors was funded by the UK Government's Department for Science, Innovation & Technology through the UK's National Measurement System programmes.

**Contributors** HC, AL, CH and JT conceived the original study idea. HC, CAP, RA, EAC, SdL, AH-S, CH, NK, DO'C, AS, SAT, JT and AL contributed to designing and planning the study. HC and AL wrote the protocol and obtained ethics approval. HC drafted the first version of the manuscript. HC, CAP, RA, EAC, SdL, AH-S, CH, NK, DO'C, AS, SAT, JT and AL contributed to writing and reviewing the manuscript.

**Funding** This research was conducted as part of a PhD studentship funded by the University of Surrey and undertaken by HC. This study received seed funding from the Faculty of Health and Medical Sciences, University of Surrey awarded to AL.

**Competing interests** JT's role in Surrey and Sussex Cancer Alliance is partly funded by Cancer Research UK.

**Patient and public involvement** Patients and/or the public were involved in the design, or conduct, or reporting or dissemination plans of this research. Refer to the Methods section for further details.

**Patient consent for publication** Not applicable.

**Provenance and peer review** Not commissioned; externally peer reviewed.

**ORCID iDs**
Hugh Claridge http://orcid.org/0000-0001-5998-2860
Simon de Lusignan http://orcid.org/0000-0001-5613-6810
Agnieszka Lemanska http://orcid.org/0000-0003-4849-2430

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
