## [Reviewer comments · BMJ Open]

ARTICLE DETAILS

TITLE (PROVISIONAL)	Determining the feasibility of calculating pancreatic cancer risk scores for people with new-onset diabetes in primary care (DEFEND PRIME): study protocol
AUTHORS	Claridge, Hugh; Price, Claire; Ali, Rofique; Cooke, Elizabeth; de Lusignan, Simon; Harvey-Sullivan, Adam; Hodges, Catherine; Khalaf, Natalia; O'Callaghan, Dean; Stunt, Ali; Thomas, Spencer; Thomson, Joanna; Lemanska, Agnieszka

VERSION 1 – REVIEW

REVIEWER	Sahoo, Jayaprakash Jawaharlal Institute of Postgraduate Medical Education
REVIEW RETURNED	23-Nov-2023

GENERAL COMMENTS	1. Mention the details about the sample size calculation. 2. Add few lines about the effect of COVID pandemic on this study.
---

REVIEWER	Gonda, Tamas A NYU
REVIEW RETURNED	25-Nov-2023

GENERAL COMMENTS	This is an important, ambitious and well-designed study that will produce impactful results for screening for pancreatic cancer Minor comments: - the abstract methods undersells or minimally describes the goal of the study. here it almost sounds like they will just be assessing the feasibility of a small amount of data extraction where as their table is much more detailed re valraibles. This section should be more clear and expanded - A1C should be given in both in % and mmol/mol - a flow sheet of enrollment and data collection as a figure would be very helpful
---

VERSION 1 – AUTHOR RESPONSE

Reviewer 1:

1a. Mention the details about the sample size calculation.

i. Thank you for highlighting this - we have now clarified the justification for not having a sample size calculation in the 'Discussion' section where we explain the chosen sample size.

1b. Add few lines about the effect of COVID pandemic on this study.

i. We have now added in more information relating to this in the 'Discussion' section.

Reviewer 2:

2a. The abstract methods undersells or minimally describes the goal of the study. Here it almost sounds like they will just be assessing the feasibility of a small amount of data extraction where as their table is much more detailed re variables. This section should be more clear and expanded

i. Thank you for flagging this - we have made adjustments to the content of the 'Methods and analysis' section of the abstract. We hope the wording changes we have made help to clarify the complexity of the study.

2b. A1C should be given in both in % and mmol/mol

i. We have added in the HbA1c equivalent % for the mmol/mol inclusion criteria mentioned, thank you for highlighting this. However, we have not added the HbA1c % to Tables 1 and 2 (data extraction and descriptive statistics tables) as we will not be extracting HbA1c % results, as mmol/mol is the unit of measurement used in the UK. We suggest that including % in these tables may cause confusion, implying the participating primary care practices will need to extract % or convert from mmol/mol to %, which is not something they will be doing.

2c. A flow sheet of enrolment and data collection as a figure would be very helpful

i. Thank you for the suggestion, we have now added Figure 1 to help demonstrate this.

We thank you all again for your input on the protocol and hope that the changes made suitably address your recommendations.